# Biological Activities of Extracts from Aerial Parts of *Salvia pachyphylla* Epling Ex Munz

**DOI:** 10.3390/plants7040105

**Published:** 2018-11-23

**Authors:** Gabriela Almada-Taylor, Laura Díaz-Rubio, Ricardo Salazar-Aranda, Noemí Waksman de Torres, Carla Uranga-Solis, José Delgadillo-Rodríguez, Marco A. Ramos, José M. Padrón, Rufina Hernández-Martínez, Iván Córdova-Guerrero

**Affiliations:** 1Facultad de Ciencias Químicas e Ingeniería, Universidad Autónoma de Baja California, Calzada Universidad 14418, 22390 Tijuana, Mexico; gmaltay@gmail.com (G.A.-T.); ldiaz26@uabc.edu.mx (L.D.-R.); mramos@uabc.edu.mx (M.A.R.); 2Departamento de Química Analítica, Facultad de Medicina, Universidad Autónoma de Nuevo León, Madero y Dr. Aguirre Pequeño, 64460 Monterrey, Mexico; salazar121212@yahoo.com.mx (R.S.-A.); nwaksman@gmail.com (N.W.d.T.); 3Centro de Investigación Científica y Educación Superior de Ensenada (CICESE), Carretera Tijuana-Ensenada 3918, 22860 Ensenada, Mexico; urangacarla@gmail.com; 4Facultad de Ciencias, Universidad Autónoma de Baja California, Carretera Tijuana-Ensenada km. 103, 22860 Ensenada, Mexico; jdelga@uabc.edu.mx; 5Instituto Universitario de Bio-Orgánica “Antonio González”, Universidad de La Laguna, Avenida Astrofísico Francisco Sánchez 2, 38206 La Laguna, Spain; jmpadron@ull.edu.es

**Keywords:** *Salvia pachyphylla*, plant extracts, antioxidant, antimicrobial, antiproliferative, enzyme inhibitory

## Abstract

The antioxidant, antimicrobial, antiproliferative, and enzyme inhibitory properties of five extracts from aerial parts of *Salvia pachyphylla* Epling ex Munz were examined to assess the prospective of this plant as a source of natural products with therapeutic potential. These properties were analyzed by performing a set of standard assays. The extract obtained with dichloromethane showed the most variety of components, as they yielded promising results in all completed assays. Furthermore, the extract obtained with ethyl acetate exhibited the greatest antioxidant activity, as well as the best xanthine oxidase inhibitory activity. Remarkably, both extracts obtained with *n*-hexane or dichloromethane revealed significant antimicrobial activity against the Gram-positive bacteria; additionally, they showed greater antiproliferative activity against three representative cell lines of the most common types of cancers in women worldwide, and against a cell line that exemplifies cancers that typically develop drug resistance. Despite that, other extracts were less active, such as the methanolic or aqueous; their results are promising for the isolation and identification of novel bioactive molecules.

## 1. Introduction

The genus *Salvia* belongs to a large family of flowering plants, Lamiaceae, which comprises about 252 genera and 7200 species [1,2]. Several species of *Salvia* are cultivated for their aromatic features and serve as flavorings, food condiments, cosmetics and perfume additives, and folk medicines [3]. Considering the latter, scrutiny of their chemical constituents have revealed the presence of a vast assortment of active compounds, some of them with antibacterial [4,5,6,7], antiviral [8,9], antitumor [10,11,12,13], antioxidant [14,15,16,17], antidiabetic [18,19], and antiparasitic [20] properties. Additionally, some species have been used etnopharmacologically for the treatment of mental and nervous illness [21], as well as for gastrointestinal conditions [22,23]. Furthermore, phytochemical studies have led to the isolation of many types of diterpenoids, such as abietane, ictexane, labdane, neoclerodane, and phenalenone [24,25,26], triterpenes and sterols [27], along with anthocyanins, coumarins, polysaccharides, flavonoids, and phenolic acids [22].

*Salvia pachyphylla* Epling ex Munz (blue sage) is a perennial herbaceous plant distributed from the state of California (USA) to the peninsula of Baja California (Mexico) [28]. The traditional medicine of Native American communities has taken advantage of the curative goods of blue sage and, currently, serves to treat flu symptoms, menstrual depression, and hysteria [29]. Several abietane diterpenoids with pharmacological properties have been isolated from the aerial parts of *S. pachyphylla* [30]. Considering the therapeutic potential of this plant, our study was directed towards identifying specific biological activities existing in different extracts from the aerial parts of *S. pachyphylla*. This approach represents the initial stage of a major survey aimed to isolate and identify phytochemicals with pharmacological potential.

Despite the recent dominance of synthetic chemistry as the foremost method to generate new or improved therapeutic agents, there is still a potential for plants to serve as a natural source of novel drugs [31]. Interestingly, the chemical diversity of natural products is complementary to the diversity found in synthetic libraries. However, natural products are sterically more complex and have a greater diversity because of their long evolutionary selection process [32].

Examples of successful medicines derived from natural products include antibiotics, enzyme inhibitors, immunosuppressive drugs, and antiparasitic agents [33]. The antitumor area is likely the greatest impact of drugs derived from plants, where vinblastine, vincristine, taxol, and camptothecin have improved the effectiveness of chemotherapy of some of the deadliest cancers.

Here, we reported the antioxidant, antimicrobial, antiproliferative, and enzyme inhibition properties of five extracts (obtained with n-hexane, dichloromethane, ethyl acetate, methanol, and water) from the aerial parts of *S. pachyphylla*. These properties were examined performing a set of standard in vitro assays.

## 2. Results

### 2.1. Antioxidant Screening

The antioxidant activity was evaluated using the β-carotene-linoleic acid assay and the 1,1-diphenyl-2-picrylhydrazyl DPPH radical-scavenging capacity assay (Figure 1 and Figure 2). In the β-carotene-linoleic test, the best activity was detected in the ethyl acetate extract (84%), immediately followed by the dichloromethane extract (83%); all five extracts showed higher activity, as compared with the reference compound α-tocopherol (8%). On the other hand, in the DPPH system, the ethyl acetate extract remained at the top in this activity, revealing an EC_50_ of 0.28 mg/mL. In addition, the extracts obtained with n-hexane or water showed similar values (0.41 and 0.51 mg/mL, respectively). Remarkably, neither of the extracts exhibited a comparative value with quercetin, the reference compound (0.003 mg/mL).

### 2.2. Antimicrobial Activity

The antimicrobial activity was examined by determining the minimum inhibitory concentrations (MIC) using five bacterial strains and three antibiotics as the reference (Table 1). Interestingly, the extracts obtained with n-hexane or dichloromethane showed significant activity against the Gram-positive *Staphylococcus aureus* and *Enterococcus faecalis*, as well as for the Gram-negative *Escherichia coli*. Furthermore, *E. coli* also exhibited considerable sensitivity to the ethyl acetate extract. Remarkably, the methanolic and the aqueous extracts were inactive against the all bacteria tested. Moreover, *Klebsiella pneumoniae* and *Acinetobacter baumannii* were insensitive to all *S. pachyphylla* extracts examined.

### 2.3. Xanthine and Acetylcholinesterase Inhibitory Assay

The enzymatic evaluation results are shown on Table 2. In the acetylcholinesterase inhibition assay, the extracts did not show a remarkable activity, only the dichloromethane extract presented a slight activity with an IC_50_ of 191.7 µg/mL; however, such a result is far away from the positive control galantamine (0.278 µg/mL). In the xanthine oxidase inhibition assay, better results were obtained, with IC_50_ values for the ethyl acetate and methanol extracts standing out with 11.7 and 19.5 µg/mL, respectively, although they did not surpass the drug allopurinol, used as control (0.842 µg/mL). The rest of the extracts did not show significant activity.

### 2.4. Antiproliferative Activity

The antiproliferative activity was obtained by measuring the concentration needed to decrease cell propagation by 50% (GI_50_) using six human cancer cell lines and three well-known anti-cancer drugs (Table 3). All extracts exhibited a degree of effectiveness against all cell lines tested. Specifically, extracts obtained with dichloromethane or n-hexane were the most active against all the evaluated cell lines, showing GI_50_ values between 5.4 and 11 µg/mL. Both extracts showed higher cytotoxicity against cell lines SW1573, T-47D, and WiDr, with concentrations of 6.6, 11, and 8.6 µg/mL and 7.7, 9.9, and 9.9 µg/mL for n-hexane and dichloromethane extracts, respectively; in both cases, the extracts surpassed the positive control etoposide (GI_50_ of 15, 22, and 23 µg/mL against SW1573, T-47D, and WiDr, respectively) and cisplatin (GI_50_ of 15 and 26 µg/mL against T-47D and WiDr, respectively).

## 3. Discussion

Oxidative stress plays a key role in the development of several pathophysiological conditions, such as neurodegenerative and cardiovascular diseases, cancer, and diabetes; natural antioxidants ingested in the daily diet protect the cells against the damage produced by an excess of reactive oxygen species (ROS) [34,35,36,37]. There are several techniques for determining the antioxidant capacity; the difference between these methods lies in the assay principle and the experimental conditions [38]. We worked with two of the most used techniques for the antiradical evaluation: DPPH^•^ and ABTS^•+^ [39], Their high popularity is due to the speed, reproducibility, and simplicity of their procedures. On the other hand, β-carotene gives information regarding the capacity of lipidic peroxidation inhibition [40]. Several studies suggested a good antioxidant potential from *Salvia* species around the world, mainly because of the presence of diterpenes, such as carnosol (1), rosmanol (2), and isorosmanol (3) (Figure 3) [41]. These three compounds are also described in a phytochemical study of *S. pachyphylla* by Guerrero et al. [30].

Cuvelier et al. [41] also made an antioxidant evaluation of isolated diterpenes from *Salvia officinalis* such as carnosol and rosmanol by the Method of Antioxidative Power (AOP test) [42], and established that the activity of all these components was related to their phenolic structure. Phenolic diterpenes are widely known to be excellent antioxidants [43,44].

In another study with abietane diterpenes isolated from a dichloromethane extract of *Salvia officinalis* L, made by Miura et al. [45] with the Oil Stability Index (OSI) method [46] and the radical scavenging activity on the DPPH radical, these diterpenes exhibited a remarkably strong antioxidant activity, which was comparable to the one of the standard α-tocopherol. The authors mention that the compounds showing strong antioxidant activity were commonly included an ortho-dihydroxy group in the molecule structure, suggesting that the antioxidant activity is due to the presence of this ortho-dihydroxy group on the C-ring. This characteristic is also found in the metabolites mentioned above that can be found in *S. pachyphylla* extracts.

In the present work, the ethyl acetate extract showed a better antioxidant activity for the two evaluated techniques, along with the dichloromethane extract in the β-carotene assay. These results are in accordance to those obtained by Şenol et al. (2010) and Loizzo et al. (2010) [47,48], where the intermediate polarity extracts (ethyl acetate) from different *Salvia* species presented better results for the same antioxidant techniques of β-carotene and the DPPH radical. The phytochemical background, described above with plants of the *Salvia* genus, suggests that the abietane-type diterpenes can be found in the intermediate polarity extracts of *S. pachyphylla* and could be the responsible for its antioxidant activity.

For the antimicrobial susceptibility analysis, one of the more employed techniques is the calculation of the minimal inhibitory concentration (MIC), which allows researchers to determine easily the minimum quantity of an analyte capable of inhibiting the visible growth of a microorganism [49]. In the antimicrobial evaluation, n-hexane and dichloromethane extracts only matched the MIC of oxacillin in the ORSA (125 µg/mL) and got near to cephalotin with the same bacteria (62.5 µg/mL). Although the reference antibiotics surpassed the extracts activities, it can be noticed that n-hexane and dichloromethane extracts presented a better activity against Gram-positive bacteria *S. aureus* and *E. faecalis*, and only with *E. coli* on the Gram-negative evaluated bacteria. From the higher-polarity extracts (ethyl acetate, methanol, and water), only ethyl acetate presented some degree of activity with ORSA and *E. coli*; however, this was still far away from the reference antibiotics. Methanol and water extracts had even higher MIC than the n-hexane and dichloromethane ones. While our extracts were less active than the references, it can be pointed out that in general, the better results were obtained from the low to medium polarity extracts, against the Gram-positive bacteria. Our results are consistent with those described by Vlietinck et al. [50], who suggested that Gram-positive bacteria are significantly more susceptible to plant-derived extracts. This may be attributed to the fact that the cell wall in Gram-positive bacteria consist of a single layer, while the Gram-negative cell wall is a multilayered and quite complex structure [51]. Previous studies suggested that the antibacterial activity from *Salvia* extracts over Gram-negative bacteria such as *E. coli* depends on the nature of the studied extract [52]. As has been mentioned before, the biological activity of the *Salvia* extracts is related to the presence of abietane diterpenes [53]; in a bioassay-guided study of *Rosmarinus officinalis* L, several extracts were evaluated against different microorganisms responsible for initiating dental caries, *Enterococcus faecalis* among them, and their MICs showed that the higher antibacterial activity was from an extract where carnosic acid (4) and carnosol (1) (Figure 3) were identified as the major compounds by HPLC analysis. As stated before, these metabolites are also found in *S. pachyphylla*.

On the other hand, Oluwatuyi et al. [54], as part of a project to characterize plant-derived natural products that modulate bacterial multidrug resistance (MDR), conducted a bioassay-guided fractionation of a chloroform extract of the aerial parts of *Rosmarinus officinalis*, leading to the characterization of the diterpenes carnosic acid (1) and carnosol (2). The antibacterial activities of these natural products make them interesting potential targets for synthesis.

The screening of natural products in the search of medically relevant enzyme inhibitors remains a viable approach for isolation of novel compounds with specific pharmacological properties, which still has a great potential for further studies. Here, two activity assays were used to identify enzyme inhibitors, within each extract, with therapeutic potential (Table 2): xanthine oxidase (XO) and acetylcholinesterase (AChE) inhibition assays. Nowadays, the potential of natural products has not yet been explored in the search of new treatments for the control of Xanthine Oxidase related diseases [55]. Xanthine oxidase catalyzes the oxidation of hypoxanthine to xanthine and uric acid; however, under certain conditions, this can generate superoxide. It has been proved that XO inhibitors can be helpful for the treatment of liver disease and gout [56]. Acetylcholinesterase catalyzes the hydrolysis of the neurotransmitter acetylcholine into choline and acetic acid [57]. Modulation of acetylcholine levels using acetylcholinesterase inhibitors is among the major strategies to address diverse neurodegenerative diseases [58]. Remarkably, extracts with a prospective inhibitory effect showed a concentration-dependent trend and IC_50_ values were estimated, with the ethyl acetate extract that exhibited a significant effect over XO, while the dichloromethane extract showed a considerable effect on AChE. Unfortunately, other extracts showed a little or poor after-effect on either of the tested enzyme activities, hence, they were considered as inactive.

Regarding the antiproliferative activity, the effects of the n-hexane and dichloromethane extracts over A2780 (ovarian carcinoma), HBL-100 (breast carcinoma), and HeLa (cervix adenocarcinoma), which represent three of the most common cancers in women worldwide, were noticeable. Remarkable results were also obtained against colorectal adenocarcinoma (WiDr), which exemplifies cell lines that typically show drug resistance [59]. Interestingly, these results are different from those previously reported by Guerrero et al. [30]. They tested pure diterpenes isolated from the aerial parts of two species, *S. clevelandi* and *S. pachyphylla*, which were less effective than our extracts. According to the results obtained, from all tested diterpenes, the most active were carnosol (1), 20-deoxocarnosol (5), and 16-hydroxycarnosol (6) (Figure 3) for five different cell lines. We thought that the main difference resides in the nature of the sample, as our results were generated using total extracts, suggesting a possible synergistic effect. Despite other extracts being less active, their results remain promising for pursuing novel molecules with cytotoxic effect.

## 4. Materials and Methods

### 4.1. Plant Material

The aerial parts (leaf, flower, and stem) of *S. pachyphylla* were collected in lands of the Sierra Juarez-Constitution National Park, Ensenada, B.C., México, at an elevation of 1630 m and coordinates of N: 32°01′41″; W: 115°56′11″ (Figure 4). Identification and authentication were carried out by Dr. José Delgadillo, and the voucher specimen (BCMEX9783) was deposited in the herbarium of the Autonomous University of Baja California, at Ensenada. Aerial parts (1.3 kg) were air-dried for a week, under shade (to reduce moisture content). The dried material was ground to fine powder and stored at 4 °C until use.

### 4.2. Preparation Extracts

Crude extracts were obtained through the classical Soxhlet method. Five different thimbles were uniformly packed, each one with 75 g of fine powder. The extraction was carried out using different solvents, one for each one of the thimbles (250 mL): n-hexane (HX), dichloromethane (DC), ethyl acetate (EA), methanol (MT), and distilled water (AQ). The extraction process was stopped until the solvent in siphon tube became colorless. Each extract was filtered and dried at 40 °C, using a rotary evaporator (Buchi Rotavapor^®^ R-215), until a solid or semi-solid residue was yielded. Each residue was separately further lyophilized to get a dry solid matter: HX = 60.3 g, DC = 7.9 g, EA = 24.2 g, MT = 9.9 g, and AQ = 14.5 g. All solids were kept in air tight bottles and stored at 4 °C until use.

### 4.3. β-Carotene-Linoleic Acid Assay

The antioxidant activity was assayed by the coupled oxidation of β-carotene and linoleic acid as described by Burda and Oleszek [60] with minor modifications. 1 mL of a β-carotene solution (0.2 mg/mL in chloroform) was added to an emulsion containing 0.018 mL of linoleic acid and 0.2 mL of Tween-20. Chloroform was removed (under a nitrogen environment), 50 mL of aerated deionized water (DO of 9.7 mg/L) was slowly added, and the mixture was vigorously agitated to form a stable emulsion. 5 mL of this emulsion was transferred to test tubes containing the corresponding sample (2 mg) of each extract. Immediately, the absorbance was measured at 470 nm (*A*_470_, zero time). All tubes were then incubated at 50 °C and *A*_470_ values were registered every 15 min for 2 h. A control without the antioxidant was prepared aside and α-tocopherol was used as a reference compound. The antioxidant activity (AA) was expressed as the percentage of inhibition of β-carotene bleaching, as compared to the control, and calculated using the following formula:*AA* (%) = [1 − (*A_S_*^0^ − *A_S_*^120^/*A_C_*^0^ − *A_C_*^120^)] × 100
where *A_S_* and *A_C_* represent the *A*_470_ value of the sample and control, respectively, and the superscript numbers denote the time of the initial and final measurement (0 and 120 min). All determinations were performed in duplicate and replicated at least three times.

### 4.4. DPPH Radical-Scavenging Capacity Assay

The radical-scavenging activity was performed as described by Burda and Oleszek [60] with slight modifications. For the evaluation of each extract, a stock solution (4 mg/mL) was prepared and serially two-fold diluted (down to 0.003 mg/mL) with methanol. An aliquot of each dilution (1 mL) was mixed with 1 mL of a methanolic solution of 1,1-diphenyl-2-picrylhydrazyl (DPPH at 0.03 mg/mL). At the same time, a control containing 1 mL of methanol and 1 mL of the DPPH solution was prepared. The mixtures were incubated at room temperature in the dark for 5 min. Using methanol as a blank, the absorbance was quantified at 517 nm (*A*_517_). The radical-scavenging activity was calculated as the percentage of DPPH decoloration using the following formula:*DPPH* (%) = [1 − (*A*/*B*) × 100]
where *A* and *B* represent the *A*_517_ value of the control and sample, respectively. All determinations were performed in duplicate and replicated at least three times. For each extract, the percentage of DPPH decoloration was plotted against the concentration of each dilution. The concentration required to decrease the absorbance of DPPH by 50% was obtained by interpolation, from a linear regression analysis, and expresses the EC_50_ value. Quercetin was used as a reference compound.

### 4.5. Antimicrobial Assay

Antibacterial activity was tested using a microdilution assay following the National Committee for Clinical Laboratory Standards (NCCLS), due its precision and reproducibility [61,62], and MIC was defined as the lowest concentration that prevents visible growth of bacteria. Six strains of microorganisms were used for antimicrobial testing including Gram-positive bacteria (*Enterococcus faecalis*, *Staphylococcus aureus*, and Oxacillin-resistant *Staphylococcus aureus*) and Gram-negative bacteria (*Acinetobacter baumannii*, *Escherichia coli*, and *Klebsiella pneumoniae*); all strains were provided by the Regional Center for Infection Diseases, School of Medicine, Autonomous University of Nuevo Leon (Monterrey, Mexico). All strains were plated on a Müeller–Hinton agar (Becton Dickinson) and incubated at 37 °C for 24 h. Four or five colony forming units were suspended in saline solution and the optical density was adjusted to the turbidity of the 0.5 McFarland Standard. Working suspensions were prepared by a 1:50 dilution in Müeller–Hinton broth. For the evaluation of each extract, a stock solution (6 mg/mL in 5% DMSO) was prepared and serially two-fold diluted with Müeller–Hinton broth in a 96-well microtiter plate (down to 0.5 μg/mL, final highest DMSO concentration 0.83%). One volume (0.1 mL) of working suspension was added to each well. The antibiotics cephalothin, oxacillin, and vancomycin were used as reference compounds. Controls without bacterial cells (medium control) and without extract or antibiotic (growth control) were prepared aside. Plates were incubated at 37 °C for 48 h and growth was visually examined.

### 4.6. Xanthine Oxidase Inhibition Assay

There are several assays employed for the Xanthine oxidase activity quantification, the more widely used is by the spectrophotometric determination of uric acid [63], the final reaction product. In this study, the inhibition of the XO activity was evaluated using the easy and sensitive protocol described by Havlik et al. [64]. A reaction solution containing 0.4 mL of 120 mM phosphate buffer (pH 7.8) and 0.33 mL of 150 mM of xanthine was supplemented with 0.25 mL of inhibitor solution (extract or reference) and mixed thoroughly. The reaction was started by adding 0.02 mL of XO enzyme solution (0.5 U/mL). After 3 min of incubation at 24 °C, the uric acid formation was determined by measuring the absorbance at 295 nm (*A*_295_). A reaction without inhibitor was used as control and allopurinol served as a reference compound. The inhibition percentage of XO activity was calculated using the following formula:*XO inhibition* (%) = [1 − (As/Ac)] ×100
where *A_S_* and *A_C_* represent the initial velocity of reactions with sample and control, respectively. All determinations were performed in duplicate and replicated at least three times. The concentration required to decrease the activity of *XO* by 50% was obtained by interpolation, from a linear regression analysis, and expresses the IC_50_ value.

### 4.7. Acetylcholinesterase (AChE) Inhibition Assay

The acetylcholinesterase inhibition activity was determined with the technique described by Adewusi et al. [65], which is a modification of the widely known and employed Ellman method [66]. For each determination, wells of a microtiter plate were filled with 25 μL of 15 mM acetylthiocholine iodide (in water), 125 μL of 3 mM DTNB in buffer C (50 mM Tris-HCl, pH 8.0, containing 0.1 M NaCl and 0.02 M MgCl_2_·6 H_2_O), 72.5 μL of buffer B (50 mM Tris-HCl, pH 8.0, containing 0.1% BSA), and 2.5 μL of inhibitor solution (extract or reference, in DMSO) and were mixed thoroughly. Absorbance was measured at 412 nm (*A*_412_) every 45 s, three times consecutively. Thereafter, 25 μL of AChE enzyme solution (0.2 U/mL) was added to each well and *A*_412_ was measured five times consecutively every 45 s. A reaction without inhibitor was used as control and galantamine served as the reference compound. Any increase in absorbance due to the spontaneous hydrolysis of the substrate was corrected by subtracting the *A*_412_ before adding the enzyme. The inhibition percentage of AChE activity was calculated using the following formula:*AChE inhibition* (%) = [1 − (As/Ac)] ×100
where *A_S_* and *A_C_* represent the initial velocity of reactions with sample and control, respectively. All determinations were performed in duplicate and replicated at least three times. The concentration required to decrease the activity of AChE by 50% was obtained by interpolation, from a sigmoidal regression analysis, and expresses the IC_50_ value.

### 4.8. Cell lines and Culture Conditions

Five human cancer cell lines were used in this study: A2780 (ovarian carcinoma), HBL-100 (breast carcinoma), HeLa (cervix adenocarcinoma), SW1573 (lung carcinoma), T-47D (breast ductal carcinoma), and WiDr (colorectal adenocarcinoma). All line cells were maintained in RPMI (Roswell Park Memorial Institute) 1640 media supplemented with 5% heat-inactivated FCS (Fetal Calf Serum) and 2 mM l-glutamine at 37 °C, 5% CO2, and 95% humidity. Exponentially growing cells were trypsinized and resuspended in medium containing 2% FCS and antibiotics (100 U/mL of penicillin G and 0.1 mg/mL of streptomycin). Single cell suspensions showing >97% viability, by trypan blue dye exclusion assay, were subsequently counted. After counting, dilutions were made to give the appropriate cell densities required for antiproliferative testing.

### 4.9. Antiproliferative Assay

Antiproliferative testing was performed using the Sulforhodamine B assay (SRB), the preferred high-throughput assay from National Cancer Institute (NCI, NIH, USA), as reported by Miranda et al. [59], with slight modifications. The SRB assay is based on the dye union to the basic amino acids of the cellular proteins, and the colorimetric evaluation provides an estimation of the total protein mass, which is related to the number of cells. This assay presents an excellent linearity, sensibility, and low costs compared to others [67]. Each extract was initially dissolved in DMSO at 400 times the desired maximum concentration to test. Six thousand cells were inoculated to each well of a microtiter plate (100 μL of a suspension of 6 × 10^4^ cells per mL). One day after plating, all testing samples and reference compounds were added to corresponding wells (triplicated). Samples were tested in six decimal serial dilutions starting at 250 µg/mL. Control cultures were tested against equivalent concentrations of DMSO (0.25% as a negative control). After 48 h of incubation at permissive conditions, cell cultures were treated with 25 μL ice-cold 50% TCA and fixed at 4 °C for 60 min. Thereafter, the SRB assay was performed. The absorbance of each well was measured at 492 nm (*A*_492_). All absorbance values were corrected for background *A*_492_ (control wells containing just culture media). As recommended by the NCI, the concentration that causes 50% growth inhibition, GI_50_ value, was corrected by count at time zero; thus, GI_50_ is the concentration where [(T − T_0_)/(C − T_0_)] = 0.5. The absorbance of the test well after 48 h is T, the absorbance at time zero is T_0_, and the absorbance of the control is C. For all these calculations, Excel spreadsheets were used. GI_50_ values were computed from the dose-response curves.

## 5. Conclusions

In conclusion, aerial parts of *S. pachyphylla* have prospective potential as a source of natural products that could act as antioxidants, antimicrobial and antiproliferative compounds, or enzyme inhibitors.

Although our findings could be the outcome of a synergistic effect, they support the notion of aiming our next approach towards the isolation and identification of novel molecules with therapeutic potential.

## Figures and Tables

**Figure 1 plants-07-00105-f001:**
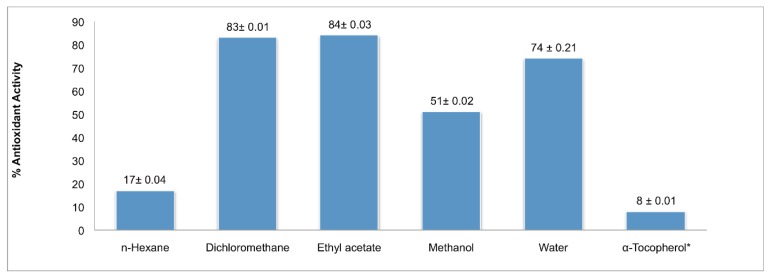
β-carotene-linoleic acid assay of extracts from aerial parts of *Salvia pachyphylla*. * Used as a reference compound. Values are mean ± SD, *n* = 3.

**Figure 2 plants-07-00105-f002:**
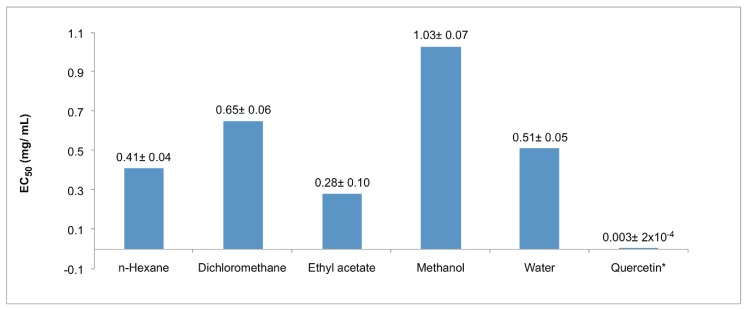
DPPH radical-scavenging capacity assay of extracts from aerial parts of *S. pachyphylla*. * Used as a reference compound. Values are mean ± SD, *n* = 3.

**Figure 3 plants-07-00105-f003:**
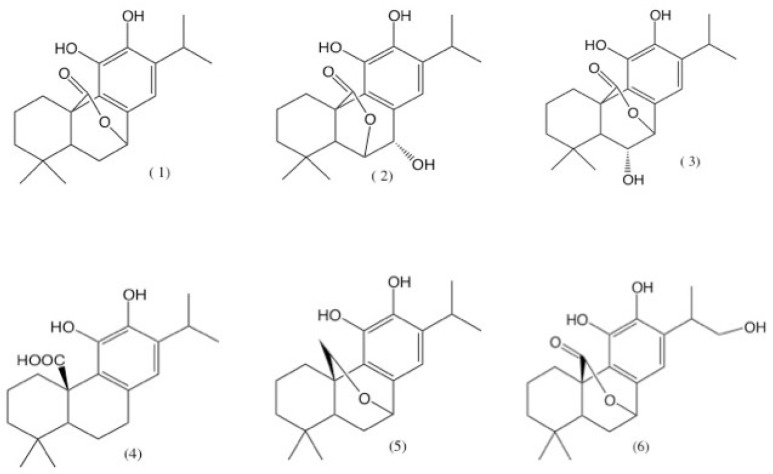
Abietanic diterpenes: carnosol (**1**), rosmanol (**2**), isorosmanol (**3**), carnosic acid (**4**), 20-deoxocarnosol (**5**), and 16-hydroxycarnosol (**6**).

**Figure 4 plants-07-00105-f004:**
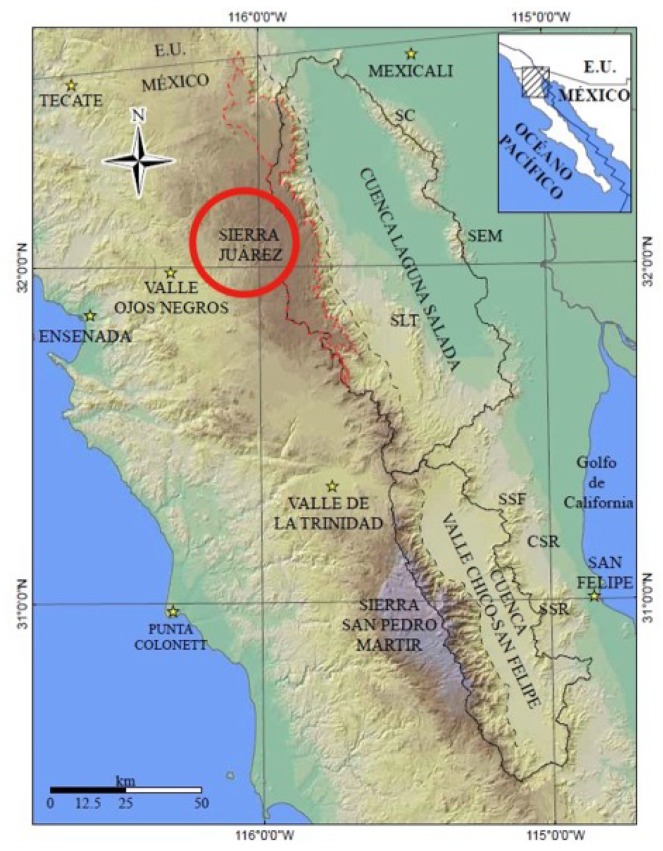
Sierra Juarez-Constitution National Park (marked by red circle), Ensenada, BC, México.

**Table 1 plants-07-00105-t001:** In vitro antimicrobial activity of extracts from aerial parts of *S. pachyphylla*.

Antimicrobial Activity (Minimum Inhibitory Concentrations (MIC), μg/mL)						
Extracts or Controls	*S. aureus*	*ORSA* ^a^	*E. faecalis*	*E. coli*	*K. pneumoniae*	*A. baumannii*
*n*-Hexane	62.5	125	250	250	>1000	>1000
Dichloromethane	62.5	125	250	250	>1000	>1000
Ethyl acetate	1000	250	>1000	250	>1000	>1000
Methanol	1000	>1000	>1000	>1000	>1000	>1000
Water	1000	>1000	>1000	>1000	>1000	>1000
Oxacillin *	0.48	125	31.2	0.487	>1000	>1000
Cephalothin *	0.48	62.5	31.2	1	>1000	62.5
Vancomycin *	0.48	1.95	1.95	>250	>1000	250

* Used as a reference compound. ^a^ Oxacillin-resistant *Staphylococcus aureus*.

**Table 2 plants-07-00105-t002:** Acetylcholinesterase (AChE) and Xanthine Oxidase (XO) inhibitory activity of the extracts from aerial parts of *S. pachyphylla*.

IC_50_ (μg/mL)		
Extracts or Controls	AChE	XO
*n*-hexane	>400	254.5 ± 31.7
Dichloromethane	191.7 ± 13.1	86 ± 6.0
Ethylacetate	314.3 ± 43.2	11.7 ± 2.4
Methanol	>400	19.5 ± 0.47
Water	>400	61.8 ± 0.9
Galantamine *	0.278 ± 0.01	ND
Allopurinol *	ND	0.842 ± 0.078

* Used as a reference compound. Values are mean ± SD, *n* = 3. ND, not determined.

**Table 3 plants-07-00105-t003:** Antiproliferative activity of extracts from the aerial parts of *S. pachyphylla*.

GI_50_ (µg/mL)						
Extracts or Controls	A2780	HBL-100	HeLa	SW1573	T-47D	WiDr
*n*-hexane	6.0	5.9	6.1	6.6	11	8.6
Dichloromethane	5.4	6.7	8.3	7.7	9.9	9.9
Ethylacetate	6.5	18	40	15	38	53
Methanol	34	64	71	70	>100	>100
Water	52	55	77	74	>100	>100
Cisplatin *	ND	1.9	2.0	3.0	15	26
Etoposide *	ND	2.3	3.0	15	22	23
Camptothecin *	ND	ND	0.6	0.25	2.0	1.8

* Used as a reference compound. ND, not determined. Human cancer cell lines: A2780, ovarian carcinoma; HBL-100, breast carcinoma; HeLa, cervix adenocarcinoma; SW1573, lung carcinoma; T-47D, breast ductal carcinoma; and WiDr, colorectal adenocarcinoma.

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
