# Peer review of "Biological Activities of Extracts from Aerial Parts of Salvia pachyphylla Epling Ex Munz"

_plants, 2018, doi:10.3390/plants7040105_

Round 1

Reviewer 1 Report

This article analyzes the biological activities of some extracts from aerial parts of Salvia pachyphylla; the experimental section is very detailed, and results are reported in a clear and concise way. Finally, Authors support the content of their paper with a rich bibliographic section. In my opinion, it is a clear and well-written piece of work and I think it deserves to be published in Plants in present form.

Author Response

Thank you for the review you gave to our manuscript, and your observations of it. Right now we are make a few adjustments to it that the other reviewers asked for, but we are confident that the manuscript will be published soon, so it will be readily available for the scientific community.

Thanks again for your support.

The authors

Reviewer 2 Report

Although the authors suggest that the published studies represents only the initial stage of a more complex study aimed to isolate and identify phytochemicals with pharmacological potential from Salvia pachyphylla, a minimum correlation between the biological activities tested and the phytochemical composition of the species is necessary (experimental data or at least some hypothesis based on the literature reported constituents of the studied plant).

Author Response

See the attachment for the responses of your observations.

Thank you for your review.

Reviewer 3 Report

The antimicrobial, antioxidant, anti-xanthine oxidase, anti-acetylcholinesterase, and anti-proliferative activities of different extracts of Salvia pachyphylla was evaluated by the authors.

The authors should give a brief introduction to explain why the antioxidant methods were those and not other ones. The same for the microorganisms and cancer cells assayed chosen. Why these ones and not other ones?

In what concerns the discussion of the best activity of some extracts against the growth of microorganisms, the authors should suggest which compounds could be responsible for such activities and not only describing that the activity depends on the nature of the studied extract. The same can be applied for the remaining activities. The discussion needs a huge improvement.

In the preparation of the extracts, how can the authors explain that 75 g of plant can provide lyophilized dry solid matter: HX = 60.3 g, DC = 7.9 g, EA = 24.2 g, MT = 9.9 g, and AQ = 14.5 g.?

In the carotene bleaching test, the authors used 2 mg of each extract that gave ~ 80% inhibition. The activity was much better than the reference (tocopherol). Was the mass of this vitamin the same of that used for samples?

For these reasons the manuscript needs major revision.

Author Response

(The authors gave the same response as above.)

Reviewer 4 Report

The complete (and correct) name of the species to be used is Salvia pachyphylla Epling ex Munz. It should be written as such in the title, abstract and the first use in the full text.

“Also, some species have served for the treatment of mental and nervous illness [21] as well as for  gastrointestinal conditions [22-23]”. This sentence is rather misleading, because Salvia species have never been a standard treatment for nervous illness or gastrointestinal conditions; they either have been used traditionally so (in the folk medicine) or were proposed based on non-clinical experiments. This should be clarified by phrasing the sentence appropriately, taken into account the data cited.

“Interestingly, the extracts obtained with n-hexane or dichloromethane showed significant activity  against  the  Gram-positive  S.  aureus  and  E.  faecalis,  as  well  as  for  the  Gram-negative  E.  coli.” This “significant activity” is considerably lower than that of the reference antibiotics, and this should be recognized explicitly in the results. There might be a few compounds, which, isolated, might have a similar effect, but as the results are reported, they are  substantially inferior to those of the relevant antibiotics (for each species).

“Oxidative stress plays a key role in the development of several pathophysiological conditions, as  neurodegenerative  and  cardiovascular  diseases,  cancer  and  diabetes”. Although this is indeed so, a few citations should be provided to support this statement.

  “Interestingly, these results are different from those previously reported by Cordova et al. [30].” Ref. no. 30 does not belong to Cordova; it should be corrected, but I am not able to evaluate how pertinent and accurate is the discussion made by the authors with respect to this reference.

Section 4.1. Plant material: please provide details on the identification process (who performed it and on what was based).

A 5% concentration level for DMSO is rather high. The usual concentration is less than 2%. The experimental description would suggest that this (relatively) high concentration did not impact negatively the control groups, but the authors should confirm that this was so.

The software used to perform the linear regression and “sigmoidal regression” analyses should be specified.

Section 4.9. Antiproliferative assay: it is not clear (not reported) how many concentrations were tested for each cell line (how many dilutions, it is only stated that the initial one was  400  times  the  desired  maximum  concentration)  and what method and software was used to compute the GI50.

Author Response

(The authors gave the same response as above.)

Round 2

Reviewer 3 Report

The authors have responded to the questions.

The manusscript can ce accepted.